# Epidemiology of Severe Fever with Thrombocytopenia Syndrome in Dogs and Cats in Taiwan

**DOI:** 10.3390/v15122338

**Published:** 2023-11-28

**Authors:** Chih-Ying Kuan, Shan-Chia Ou, Chao-Chin Chang, Pei-Ling Kao, Ruei-Sheng Tsai, Porjai Rattanapanadda, Tsai-Lu Lin, Ken Maeda, Tsun-Li Cheng, Ya-Jane Lee, Shih-Te Chuang, Shiun-Long Lin, Hsien-Yueh Liu, Fong-Yuan Lin, Jen-Wei Lin, Wei-Li Hsu, Chi-Chung Chou

**Affiliations:** 1Graduate Institute of Microbiology and Public Health, College of Veterinary Medicine, National Chung Hsing University, Taichung 40227, Taiwan; co99020030@email.nchu.edu.tw (C.-Y.K.); scou@dragon.nchu.edu.tw (S.-C.O.); changcc@dragon.nchu.edu.tw (C.-C.C.); m166108@gmail.com (R.-S.T.); 2Graduate Institute of Veterinary Pathobiology, College of Veterinary Medicine, National Chung Hsing University, Taichung 40227, Taiwan; gilliankao0320@gmail.com; 3Food and Drug Administration, Ministry of Public Health, Nonthaburi 11000, Thailand; p.rattanapanadda@gmail.com; 4New Taipei City Government Animal Protection and Health Inspection Office, New Taipei City 220066, Taiwan; steve.lin308@gmail.com; 5National Institute of Infectious Disease, Tokyo 162-8640, Japan; kmaeda@nih.go.jp; 6Veterinary Medical Teaching Hospital, College of Veterinary Medicine, National Chung Hsing University, Taichung 40227, Taiwan; flynstm@gmail.com; 7Veterinary Hospital, College of Bio-Resources and Agriculture, National Taiwan University, Taipei 10617, Taiwan; yajanelee@ntu.edu.tw; 8Department of Veterinary Medicine, College of Veterinary Medicine, National Chung Hsing University, Taichung 40227, Taiwan; stchuang@dragon.nchu.edu.tw (S.-T.C.); sllin1@dragon.nchu.edu.tw (S.-L.L.); 9Department of Animal Healthcare, Hungkuang University, Taichung 433304, Taiwan; lhy_vet@hk.edu.tw (H.-Y.L.); jenweilin@sunrise.hk.edu.tw (J.-W.L.); 10The iEGG and Animal Biotechnology Center, National Chung Hsing University, Taichung 40227, Taiwan

**Keywords:** Severe Fever with Thrombocytopenia Syndrome (SFTS), emerging tick-borne disease, dog, cat, RNA prevalence

## Abstract

Severe Fever with Thrombocytopenia Syndrome (SFTS), caused by the SFTS Virus (SFTSV), is a global health threat. SFTSV in Taiwan has only been reported in ruminants and wild animals. Thus, we aimed to investigate the infection statuses of dogs and cats, the animals with closer human interactions. Overall, the SFTSV RNA prevalence was 23% (170/735), with dogs showing a 25.9% (111/429) prevalence and cats at 19.3% (59/306) prevalence. Noticeably, the prevalence in stray animals (39.8% 77/193) was significantly higher than in domesticated ones (17.2%, 93/542). Among the four categories analyzed, the highest SFTSV prevalence was found in the stray dogs at 53.9% (120/193), significantly higher than the 24.2% prevalence noted in stray cats. In contrast, domesticated animals exhibited similar prevalence rates, with 17.1% for dogs and 17.2% for cats. It is noteworthy that in the domesticated animal groups, a significantly elevated prevalence (45%, 9/20) was observed among cats exhibiting thrombocytopenia compared to those platelet counts in the reference range (4.8%, 1/21). The high infection rate in stray animals, especially stray dogs, indicated that exposure to various outdoor environments influences the prevalence of infections. Given the higher human interaction with dogs and cats, there is a need for proactive measures to reduce the risk associated with the infection of SFTSV in both animals and humans.

## 1. Introduction

Severe Fever with Thrombocytopenia Syndrome (SFTS) is an emerging infectious disease that was noted as one of the most severe infectious diseases by the World Health Organization in 2017 [1]. It is transmitted by ticks, and the major tick species carrying the SFTS virus (SFTSV) include *Haemaphysalis longicornis*, *Amblyomma testudinarium*, *Ixodes nipponensis*, and *Rhipicephalus microplus* [2]. The first occurrence of SFTS took place in China in 2009, with rapid dissemination through various provinces in the country’s central, eastern, and northeastern areas [3]. Subsequently, cases of SFTS were documented in Japan and Korea in 2012 [4,5] and, more recently, in Vietnam, Taiwan, Myanmar, and Pakistan [6,7,8,9].

While tick bites are the primary way of transmitting SFTSV, certain studies have demonstrated that transmission from animals to humans can also happen in certain situations, such as aerosol use or close contact with infected cat blood or other body fluids, as well as bites from SFTSV-infected companion animals [10,11,12]. Moreover, SFTSV can spread from human to human in multiple ways, for instance, through direct contact with the blood or bloody secretions of an infected patient [13,14], transmission within the patient’s room and its immediate vicinity [15], and transmission via the ocular route [16]. Emerging viruses often propagate and intensify through two primary mechanisms: the proximity of and interactions between wildlife or non-wild animals and human communities, as exemplified by instances like the transmission of diseases from cats to humans, and the geographical expansion of key hematophagous arthropod vectors or their host animals, such as migratory birds transporting ticks to areas outside their original endemic zones [17,18]. 

Currently, vaccines or specific antiviral drugs are unavailable for SFTS infection, whether in humans or animals [19], and the treatment is mainly symptomatic. Many studies have been carried out to determine the existence of SFTSV in different animal species, including domestic animals (sheep, cattle, dogs, pigs, chickens, and rodents) and wildlife (cheetahs) [20,21,22,23]. Antibodies against SFTSV have also been confirmed in several kinds of domestic and wild animals [24,25,26,27,28], which play a pivotal role in both sustaining the life cycle of SFTSV and its sporadic transmission to humans [29,30].

Both dogs and cats are companion animals that often share the same living quarters as humans, and they have the potential to transmit SFTSV to both humans and other animals. In Korea, SFTSV has been reported to transmit from infected dogs to the owners via tick infestation [31], as well as from tick bites to an 8-year-old girl [32]. Via intramuscular inoculation, SFTSV infection was proven capable of inducing clinical symptoms in dogs; thus, SFTSV transmission could occur via contact with affected dogs. [19]. Moreover, in Japan, SFTSV was transmitted from cats to veterinary personnel through nosocomial infection [10,12] and sick cats biting their owners [11]. Cats exhibit a higher susceptibility to SFTSV compared to dogs, with a case fatality rate of 62.5% [33]. In contrast, dogs have shown varying SFTSV mortality rates, ranging from 0 [34,35,36] to 43% [37]. Consequently, understanding the epidemiological characteristics of SFTSV in companion animals is crucial for public health. Until now, there has been no comprehensive national report on the seroprevalence of SFTSV in small animals, specifically dogs and cats in Taiwan. To address this gap, the present study aimed to assess the prevalence of SFTSV-specific RNA among companion animals in Taiwan. Additionally, this study explored various factors that could potentially contribute to viral transmission. These factors encompassed geographic distribution and variations in habitat environments, specifically focusing on domesticated and stray animal populations.

## 2. Materials and Methods

### 2.1. Sample Collection

A total of 735 plasma and serum samples were enrolled in this study, with 429 samples from dogs and 306 samples from cats dating from February 2022 to June 2023. Among the dog samples, 327 were obtained from veterinary hospital patients, and 102 were obtained from stray dogs from animal shelters. Similarly, in the cat group, 215 samples were collected from veterinary clinics, and 91 samples were collected from stray cats through the Trap–Neuter–Return (TNR) program. Of note, among animal samples provided by veterinary clinics, only one group of domesticated animals with thrombocytopenia (*n* = 62) was specifically targeted, while the rest of the animals had no significant clinical signs. Ethical considerations were observed throughout the sample collection and experimental procedures, with the approval of the Institutional Animal Care and Use Committee of National Chung Hsing University (IACUC number: 109-060R). Blood samples were collected in the tubes with K2-EDTA.

### 2.2. Amplification of Viral RNA via Real-Time Reverse Transcription Polymerase Chain Reaction (qRT-PCR)

The detection of SFTSV RNA and determination of the sequences of amplicons followed one of the previous reports [38]. In brief, the isolation of total nucleic acid from animal serum (or plasma) samples was carried out using the Maxwell^®^ RSC Viral Total Nucleic Acid Purification Kit (Promega, Madison, WI, USA). The concentration of nucleic acid was measured via a NanoDrop 2000 spectrophotometer (Thermo Fisher Scientific, Waltham, MA, USA). Subsequently, reverse transcription (RT) of 0.5 μg of total RNA was performed using the iTaq™ Universal SYBR^®^ Green One-Step kit (Bio-Rad, Hercules, CA, USA) with primers (SFTSV-SF: 5′-ACCTCTTWGACCCTGAGTTWGACA-3′, SFTSV-SR: 5′-CTRAAGGAGACAGGTGGAGATGA-3′). The amplification process involved initial incubation at 42 °C for 30 min, followed by denaturation at 95 °C for 30 s and the following subsequent amplification cycles: 95 °C for 30 s and 60 °C for 60 s, repeated for 40 cycles. A final melting curve program ranging from 68 °C to 95 °C was executed. The designation of a positive case relied on achieving a Cq value of less than 38 and was further substantiated through the analysis of the melting curve. Moreover, the amplicons from each sample were resolved via electrophoresis, which was carried out at 100 V for 20 min using a 1.5% agarose gel in Tris-acetate-EDTA buffer; PCR products with the expected size were regarded as positive samples. 

For those samples displaying strong positive results for SFTSV (e.g., Cq value < 33), the partial S segment sequence was amplified through a two-step RT-nested PCR approach. Initially, cDNA synthesis was performed using superscript IV (Invitrogen, Carlsbad, CA, USA), which then served as the template for the nested PCR. The first amplification phase employed outer set primers (1F: 5′-CATCATTGTCTTTGCCCTGA-3′, 1R: 5′-AGAAGACAGAGTTCACAGCA-3′), while 2 μL of the initial PCR amplicon was utilized as the template for the subsequent amplification in the second run, utilizing inner set primers (2F: 5′-AAY AAG ATC GTC AAG GCA TCA-3′, 2R: 5′-TCA TTG TCT TTG CCC TCA AC-3′). The PCR procedure encompassed an initial denaturation at 95 °C for 5 min, followed by 40 cycles of denaturation at 95 °C for 30 s, annealing at 60 °C for 45 s, and extension at 72 °C for 1 min. The resulting product derived from nested PCR was resolved using 1.5% agarose gel. Amplicons with the expected size were then purified from the gel using the PureLink™ Quick Gel Extraction Kit (Thermo Fisher Scientific, Waltham, MA, USA). The identity of the PCR product at high quantities (>0.1 μg/μL) was further confirmed through automated sequencing conducted by Mission Biotechnology (Taipei, Taiwan).

### 2.3. Platelet Count Evaluation

A total of 137 blood samples were analyzed using an IDEXX ProCyte Dx^TM^ Hematology Analyzer (IDEXX Laboratories, Westbrook, MA, USA) for the evaluation of the platelet count. The value of platelet counts lower than the reference range recommended by the IDEXX ProCyte Dx^TM^ Hematology Analyzer were confirmed using the smeared slide before being interpreted as true thrombocytopenia. A total of 137 blood samples were analyzed and interpreted in the same manner.

### 2.4. Statistical Analysis

The correlations between viral RNA positivity and diverse risk variables, including animal species, geographical regions, and prevalence across different habitats (i.e., domesticated or stray), were assessed using a chi-squared test (IBM SPSS Statistics for Windows, Version 20.0). The platelet count variations between animals testing positive and negative for SFTSV RNA were assessed through Welch’s *t*-test using GraphPad Prism version 9.5.1 for Windows.

## 3. Results

### 3.1. Prevalence of SFTSV RNA in Dogs and Cats

The infection statuses of SFTSV in the canine and feline populations in Taiwan have yet to be determined. To address this gap, we initiated a nationwide surveillance effort aimed at uncovering the prevalence of SFTSV. The sample distribution and corresponding collection locations are presented in Table 1. A total of 735 serum (or plasma) samples were collected from 9 cities, along with a terrestrial island (Kinmen). The results indicated the presence of SFTSV RNA in 170 of these samples, constituting a prevalence of 23%. The identities of positive amplicons with concentrations higher than 0.1 μg/μL after purification were further confirmed, and the overall sequences of the partial SFTSV NP (305 bp) derived from dogs and cats exhibited high similarities, ranging from 99.3% to 100%. These sequences were closely associated with strains isolated in Taiwan (Appendix A). 

The serum samples included 429 canine and 306 feline samples. The overall prevalence of SFTSV in dogs (25.9%, 111/429) was significantly higher than that in cats (19.3%, 59/306) (*p* = 0.037).

### 3.2. Prevalence of SFTSV RNA in Domesticated and Stray Animals

As shown in Table 2, of the 735 samples in total, 542 were derived from domesticated animals obtained from veterinary clinics and 193 were derived from stray animals. Overall, 93 domesticated samples were SFTSV RNA positive (17.2%, 93/542), which is significantly lower than that in stray animals (39.9%, 77/193) (*p*-value < 0.001). The domesticated animal group comprised 327 dogs and 215 cats. Analyzing both the animal species and their respective habitats revealed that the prevalence of RNA is similar between domesticated dogs (17.1%, 56/327) and cats (17.2%, 37/215). In contrast, the SFTSV RNA prevalence in stray dogs (53.9%, 55/102) was significantly higher than that in stray cats (24.2%, 22/91) (*p* < 0.001). 

Subsequently, an assessment of the prevalence between stray and domesticated animals in each species (dog or cat) was carried out. Overall, stray dogs (53.9%, 55/102) exhibited a significantly higher prevalence compared to domesticated dogs (17.1%, 56/327). On the other hand, the SFTSV RNA prevalence was statistically no different (*p* = 0.158) between domesticated cats (17.2%, 37/215) and stray cats (24.2%, 22/91).

### 3.3. The Geographical Distribution of SFTSV in Dogs and Cats

Given the transmissible nature of SFTSV and the relatively confined area of Taiwan (covering 36,194 km^2^), a comprehensive grouping strategy was employed for the nine cities/counties where the samples were collected. These cities were categorized into four geographical regions, namely the northern, central, southern, and eastern regions, alongside the inclusion of a fifth western island (Kinmen) (Figure 1). This systematic approach allowed us to explore potential geographical correlations with SFTSV prevalence.

The sample numbers, based on this regional classification, are illustrated in Table 3. Significant variations in SFTSV RNA prevalence were identified across the five specified regions (*p* < 0.001). The highest prevalence of SFTSV was in southern Taiwan (36%), followed by central (29%), eastern (23%) and northern (18%) Taiwan and Kinmen (6%) in descending order (Table 3). It is worth noting that the counties with the highest prevalence were Yunlin (46.9%) and Tainan (38.4%), where stray animal samples were collected. As indicated above, stray animals exhibited a substantially higher prevalence rate (39.9%) in comparison to domesticated animals (17.1%). A marked discrepancy in SFTSV prevalence was observed among stray animal samples collected from various regions, and the highest SFTSV prevalence was detected in southern Taiwan (60.6%), followed by central (43.3%), northern (35.3%) and eastern Taiwan (33.3%) and the western island (8.3%) in descending sequence (Table 3). Conversely, out of the 542 samples from domesticated animals, a predominant portion was gathered from the northern region (*n* = 310). The distribution of SFTSV prevalence in domesticated animals also exhibited significant regional differences. However, unlike in the stray animals, the lowest SFTSV prevalence was identified in samples collected from veterinary clinics in southern Taiwan (7.1%).

### 3.4. The SFTSV Prevalence in Animals with Thrombocytopenia

Thrombocytopenia is one of the main features of SFTS patients [3]. In this study, a total of 137 domesticated animals, including 96 canine and 41 felines, had available information regarding platelet counts. Among these, 78 (58 dogs and 20 cats) displayed platelet counts below the reference range (dog: 148–484 k/μL; cat: 151–600 k/μL). The SFTSV RNA prevalence in animals with thrombocytopenia was 25.6% (20/78), which is higher than the prevalence (20.3%, 12/59) in animals with normal platelet counts, though without statistical significance (*p* = 0.47). Of note, the RNA detection rate was 45% (9/20) in cats with platelets below the reference range, which is significantly higher than that the cats without thrombocytopenia (4.8%, 1/21) (Table 4). On the other hand, the mean platelet count in cats that tested positive for SFTSV was significantly lower than that in SFTSV-negative cats (Figure 2). However, this phenomenon was not observed in dog samples.

## 4. Discussion

SFTS epidemics have occurred in several East Asian countries. Similar to what has been observed in humans, the occurrence of highly fatal cases of SFTSV in cats and dogs has raised the issue of veterinarians and pet owners potentially being exposed to a potential risk of contracting SFTSV infection through direct contact with infected animals. SFTSV has been emerging in Taiwan since 2019 [7]; however, its impact on small animals remains unknown. Focusing on the dog and cat populations in Taiwan, the comprehensive surveillance conducted in this study revealed a substantial viral RNA prevalence rate of 23%. It is important to underline that this high RNA carriage rate can be attributed to the presence of stray animals, constituting 39.8% of the cases. In addition, within the domesticated animal group, a notably elevated prevalence among animals exhibiting thrombocytopenia was observed, particularly in cats. 

It is well-documented that SFTSV affects a broad range of hosts, including humans, domestic animals, and wildlife [24]. While SFTSV can lead to severe illness in humans, the infection in most vertebrate animals does not result in significant clinical symptoms [39]. Nevertheless, both surveillance and experimental evidence have shown that infection with SFTSV in naturally infected captive cheetahs [23] and dogs [37], as well as in experimentally inoculated cats [39], can lead to the development of severe symptoms and even prove lethal like human cases. To date, varying rates of SFTSV seropositivity in dogs have been reported in different Asian countries. In China, seropositive rates range from 7.4% and 28.7% to 37.9%, increasing to 68.2% in regions such as Shandong [20,40], Jiangsu [41], and Henan provinces [42]. In the same context, positive rates for SFTSV RNA have been noted as 5.3% (19/359) for domesticated dogs in China [20] and 2.9% (3/103) for military dogs in a training camp in South Korea. It was reported that cats are highly susceptible to SFTSV. Research in South Korea showed that the RNA detection rate was 17.46% in feral cats [43], while it was 0.47% in confined-range cats [44]. However, a heavy infection rate of SFTSV in cats was noted in Nagasaki, Japan; in total, 33.1% (44/133) of SFTSV-suspected cats were confirmed to be infected with SFTS [45]. Importantly, apart from their diverse geographical origins, the variations in sampling approaches across these three studies may account for the differing infection rates observed. The most notable prevalence was recorded in SFTSV-suspected cats from Japan, and the suspected cases were defined as those with clinical signs such as fever, leukopenia, and thrombocytopenia. Consistently, as shown in Table 4, in the domesticated animals with thrombocytopenia representing SFTSV suspected groups, we detected the RNA sequence of SFTSV in 20 out of the 78 samples (25.6%), including 19% (11/58) and 45% (9/20) of dogs and cats, respectively. Interestingly, the increased SFTSV prevalence in thrombocytopenic animals was observed in cats but not in dogs (Table 4). Indeed, several studies have indicated that cats are highly susceptible to SFTSV infection [39,45], while dogs experimentally infected with SFTSV only showed minor clinical signs [19]. Consistent with our findings, Park et al., indicated that cats that were experimentally inoculated with SFTSV experienced decreases in platelet counts. This decrease was particularly severe in fatal cases [39]. Moreover, another study demonstrated lower levels of white blood cell and platelet counts (below 100 counts/μL) in SFTSV-suspected cats from animal hospitals [45]. A reduction in platelet counts has also been documented as a clinical manifestation in SFTSV-affected dogs, with this phenomenon being noted in 45.5% (5 out of 11) of the observed cases [36]. This correlation with thrombocytopenia might be attributed, at least in part, to the sampling strategies in those studies. In the Han et al.’s study [36], dogs that were exposed to ticks or displayed clinical symptoms resembling those of SFTS were included. However, in our current study, thrombocytopenia was the only selection criterion. The observed lower platelet counts, therefore, might potentially stem from factors other than the SFTSV infection. Additional clarification is imperative to address this contentious finding. 

Moreover, the diversity in SFTSV prevalence among feline samples from South Korea suggests that feral cats living in highly urbanized habitats likely demonstrated a higher detection rate, in contrast to those confined to specific environments [43,44]. Similarly, in our study, the prevalence among stray animals (39.8%) is significantly greater than that in domesticated animals (17.2%). Out of the five regions, the southern area exhibited the highest detection rate across the entire sample groups (36%, 53/149), as well as within the subset of stray animals (60.6%, 48/79). (Table 3). In particular, Yunlin (46.9%) and Tainan (38.4%) are the two cities with the highest infection rates (Table 1). It is noteworthy that the predominant portion of samples in these two cities consisted of stray animals (79 samples) collected via the Trap–Neuter–Return (TNR) program or from animal shelters, resulting in a positive rate of 60.6%, which is much higher than those of the domesticated groups (17.2%). This phenomenon was supported by a systemic review [24]. A meta-analysis found that the pooled seroprevalence of SFTSV was notably higher, reaching 55.20% (95% CI 31.90–78.50%) for free-ranging dogs and 6.40% (95% CI 4.30–8.60%) for confined-range dogs. It indicates that free-ranging or feral animals are more susceptible to SFTSV infection, likely due to their greater opportunity for exposure to tick bites. 

High SFTSV prevalence in stray animals may be attributed to frequent animal interactions within a community, coupled with an increased likelihood of tick infestations. Known as a tick-borne disease, SFTSV is primarily transmitted by *Haemaphysalis longicornis*, according to reports from other countries [46]. Recently, nationwide surveillance revealed that SFTSV RNA was detected in 27.7% of tick pools collected from ruminants and wildlife in Taiwan [38], and *Rhipicephalus microplus* is the predominant tick species carrying SFTSV, while other species such as *Haemaphysalis hystricis* and *Amblyomma testudinarium* collected from wildlife (boars) tested negative. However, the limitation of this study is that only serum samples were collected from animals, given the consistent application of tick control interventions for domesticated companion animals and stray animals housed in shelters. Furthermore, given the unidentified viral strain in dogs and cats, as for surveillance, primers were specifically designed to target the highly conserved region in the NP gene. Overall, the sequences among our local strains exhibited a high similarity, ranging from 99.3% to 100%. However, acquiring longer reads and targeting genes with variable sequences would offer further insights into the viral strains circulating in Taiwan. Overall, this analysis suggests close phylogenetic relationships between the sequences identified in cats and dogs.

In conclusion, this study effectively addressed the knowledge gap pertaining to the prevalence of SFTSV in small animals in Taiwan. The presence of SFTSV RNA was observed in both companion and stray dogs and cats. Notably, the prevalence of RNA was higher in stray animals, particularly those with thrombocytopenia. Given the significant mortality associated with SFTSV, special attention should be paid to animals (particularly cats) displaying thrombocytopenia in clinical settings.

## Figures and Tables

**Figure 1 viruses-15-02338-f001:**
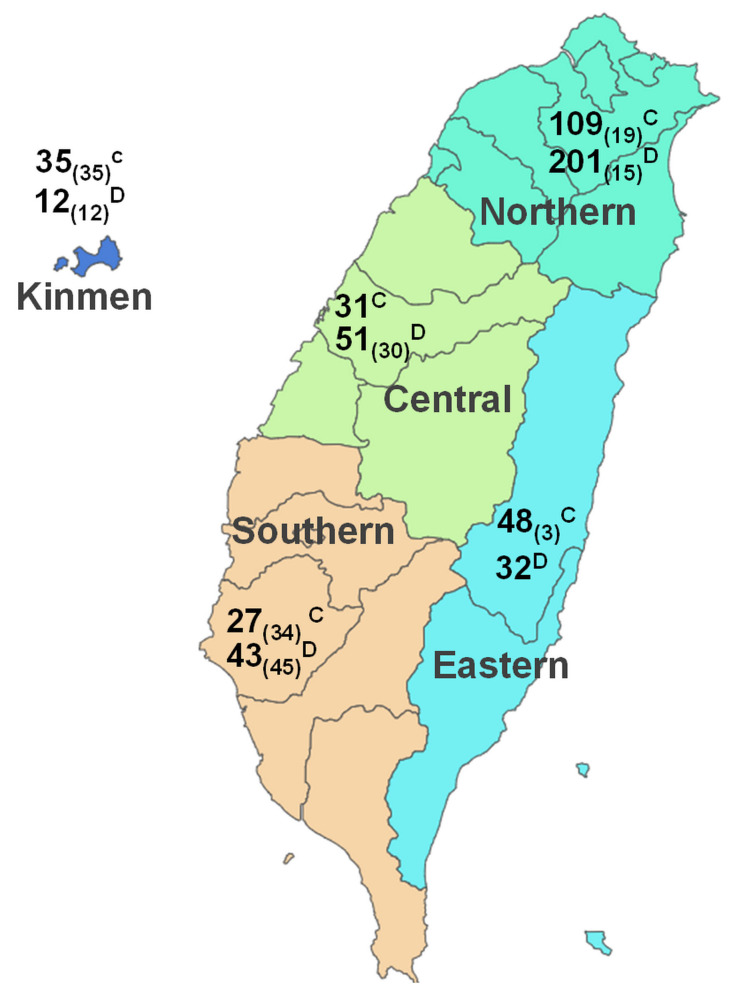
The major regions and number of animals enrolled in the study of SFTSV viral RNA detection. In total, 735 serum samples were collected from animals in Taiwan. The labels with superscript letters C or D denote the count of cat or dog samples derived from veterinary clinics, while the number of samples from stray animals is indicated within the subscript brackets.

**Figure 2 viruses-15-02338-f002:**
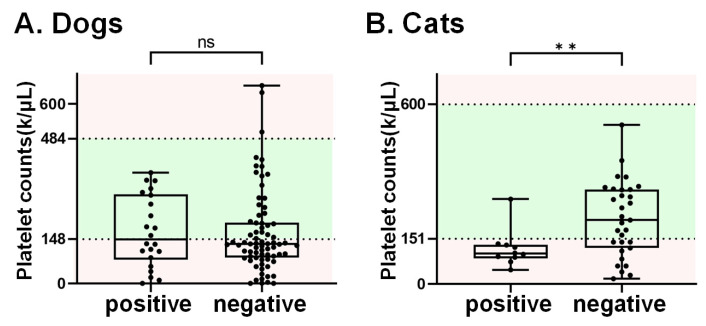
The platelet counts in dogs and cats. The 137 serum samples with records of platelet counts included 96 canine and 41 feline samples. The normal range for platelet counts is highlighted in green. Notably, the average platelet counts in cats that tested positive for SFTSV RNA (referred to as the “positive group”) were found to be significantly lower than those in cats that tested negative for SFTSV RNA. Note. ns indicates no significance, ** refers *p*-value < x0.01.

**Table 1 viruses-15-02338-t001:** The SFTSV prevalence of dogs and cats in each city.

Region (Prevalence)	City/County (Prevalence)	Animal	Total	SFTSV+	Prevalence
**Northern** (18%)	New Taipei City (35.3%)	Cat	19	6	32%
		Dog	15	6	47%
	Taipei (1.7%)	Cat	30	1	3%
		Dog	30	0	0%
	Hsinchu (0%)	Cat	37	0	0%
		Dog	28	0	0%
	Taoyuan (27%)	Cat	42	13	31%
		Dog	143	37	26%
**Central** (29%)	Taichung (28.6%)	Cat	31	7	23%
		Dog	81	25	31%
**Southern** (36%)	Yunlin (46.9%)	Cat	34	13	38%
		Dog	15	10	67%
	Tainan (38.4%)	Cat	18	2	11%
		Dog	55	26	47%
	Kaohsiung (7.4%)	Cat	9	1	11%
		Dog	18	1	6%
**Eastern** (23%)	Taitung (22.9%)	Cat	51	14	25%
		Dog	32	5	16%
**Island (West)** (6%)	Kinmen (6.4%)	Cat	35	2	6%
		Dog	12	1	8%
Summary			735	170	23%

**Table 2 viruses-15-02338-t002:** Prevalence of SFTSV in domesticated and stray animals.

**Samples ** **(Number, RNA Positive Rate)**	**Animals (735, 23.1%) ^a^**			
	**Domestic (542, 17.2%) ^a^**		**Stray (193, 39.8%) ^a^**	
	dog	cat	dog	cat
	56/327 (17.1%)	37/215 (17.2%)	55/102 (53.9%)	22/91 (24.2%)
*p*-values ^b^	0.98		<0.001	

^a^ total sample size and SFTSV RNA prevalence in the group. ^b^
*p*-values indicate the statistical differences between groups.

**Table 3 viruses-15-02338-t003:** The major geographical distributions of SFTSV prevalence in dogs and cats.

Region	Stray (193, 39.9%) ^a^			Domesticated (542, 17.2%)		
	Dog 53.4% (55/103)	Cat 24.2% (22/91)	Dog + Cat 39.9% (77/193)	Dog 17.1% (56/327)	Cat 17.2% (37/215)	Dog + Cat 17.2% (93/542)
**Northern** 18% (63/344)	6/15 (40%)	6/19 (31.6%)	12/34 (35.3%)	37/201 (18.4%)	14/109 (12.8%)	51/310 (16.5%)
**Central** 29% (32/112)	13/30 (43.3%)	-	13/30 (43.3%)	12/51 (23.5%)	7/31 (22.6%)	19/82 (23.2%)
**Southern** 36% (53/149)	35/45 (77.8%)	13/34 (38.2%)	48/79 (60.6%)	2/43 (4.7%)	3/27 (11.1%)	5/70 (7.1%)
**Eastern** 23% (19/83)	-	1/3 (33.3%)	1/3 (33.3%)	5/32 (15.6%)	13/48 (27.1%)	18/80 (22.5%)
**Kinmen** 6% (3/47)	1/12 (8.3%)	2/35 (5.7%)	3/47 (6%)	-	-	-
*p*-values ^b^	<0.05	<0.05	<0.05	>0.05	>0.05	<0.05

^a^ total sample size and SFTSV RNA prevalence in the group. ^b^ *p*-values indicate the statistical differences between groups.

**Table 4 viruses-15-02338-t004:** The SFTSV prevalence in animals with thrombocytopenia.

SFTSV Prevalence	Group (Dog + Cat) *n* = 137	Dog *n* = 96	Cat *n* = 41
**total**	23.4% (32/137)	22.9%(22/96)	24.3% (10/41)
**thrombocytopenia**	25.6% (20/78)	19% (11/58)	45% (9/20)
**reference range**	20.3% (12/59)	28.9% (11/38)	4.8% (1/21)
*p*-values ^a^	0.47	0.26	0.003

^a^ *p*-value indicates the statistical difference in prevalence between animals with thrombocytopenia and those within normal ranges.

## Data Availability

Data are contained within the article and Appendix A.

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
