# Peer review of "Epidemiology of Severe Fever with Thrombocytopenia Syndrome in Dogs and Cats in Taiwan"

_viruses, 2023, doi:10.3390/v15122338_

Round 1

Reviewer 1 Report

Comments and Suggestions for Authors

Dear authors,

Your manuscript “Epidemiology of Severe Fever with Thrombocytopenia Syndrome in canines and felines in Taiwan” describes Severe Fever with Thrombocytopenia Syndrome Virus (SFTSV) in dogs and cats from Taiwan.

The manuscript is written very well, I really like the logic of the presentation and it is clear.

My main comments concern the Materials and methods section – some methods need more accurate description. Please, revise it:

Lane 102 – “identification of SFTSV RNA sequences” – sequences could not be identified, but determined. Correct, please.

Lane 105 – “0.5 μg of total RNA” – how did you measured the quantity of total RNAs?

Lanes 113-114 – “and confirmed by gel electrophoresis.” – please, give more information about gel electrophoresis.

Lane 125 – “The resulting PCR product was then isolated” – what method/kit did you use for PCR product isolation?

Lane 126 – “the identity of the generated amplicon was confirmed through automated sequencing” – identity with what target? Or identity between all PCR products? Or something else? Please, explain more accurately.

Lane 139 – “with or without carrying SFTSV RNA” – “carrying” here is not a very appropriate word. Please rephrase.

One more remark – as indicated, you sequenced PCR fragments of the pathogen's genome, but this data is not provided anywhere. I think that would make sense. Moreover, were the determined sequences 100% identical, or was there variability observed?

Comments on the Quality of English Language

The correction of some terms is needed.

Author Response

Dear Editor and reviewers,                                            Nov, 16, 2023

The file submitted here is a revised manuscript entitled “Epidemiology of Severe Fever with Thrombocytopenia Syndrome in dogs and cats in Taiwan” for your consideration to be published in the Journal “Viruses”.

Thank you very much for this opportunity to revise our manuscript, we sincerely appreciate the time the reviewers have taken to review and the very informative and insightful comments. We hope that the revised version is deemed acceptable by Viruses. The point-to-point responses to the comments are as follows, all changes are highlighted in the revised manuscript for your convenience in review. In brief, we have added more information in Methodology, one supplementary figure, and several issues were included in the discussion as suggested by reviewers. The overall word count exceeds 4000 words.

Should you have any questions regarding this manuscript, please feel free to contact me using the information below. I wish to thank you in advance for your considerate attention to this manuscript and look forward to your reply.

Yours sincerely,

Chi-Chung Chou D.V.M. Ph.D.Distinguished Professor College/Department of Veterinary MedicineNational Chung-Hsing University145 Xing Da Rd. Taichung, TaiwanTEL: +886-4-22840895 ext. 519Email: ccchou@nchu.edu.tw

Wei-Li Hsu D.V.M. Ph.D.Distinguished ProfessorGraduate Institute of Microbiology and Public Health National Chung-Hsing University145 Xing Da Rd. Taichung, TaiwanTEL: +886 4 22840695Email: wlhsu@dragon.nchu.edu.tw

Reviewer 2 Report

Comments and Suggestions for Authors

This is a well-written and highly relevant manuscript describing SFTSV RNA prevalence in dogs and cats in Taiwan. However, there are several modifications that would improve this manuscript as follows:

Major comments:

Why was serology not performed on these animals? Seems like a missed opportunity to link evidence of infection with serostatus, and given the high RNA recovery, it seems likely that many more animals may have been exposed and seroconverted than were actively infected. I strongly suggest performing serology on these samples if they are still available.

Material and Methods- Sample Collection: Why were these animals being seen? Were these routine samples collected from healthy animals or was there inclusion criteria for collecting samples for the study? If so, what were those inclusion criteria? Considerably more information is needed regarding the source of these samples and the health status of the animals included in the study.

Discussion: There is no information on the tick vectors and their distribution on Taiwan, this information should be included as it’s epidemiologically relevant to the exposure in animals, particularly regarding the geographic distribution of cases between regions.

Minor comments:

Title: Suggest replacing “canines and felines” with “dogs and cats”- this study is only describing those species and does not address other canines or felines at all.

Line 73-74: What does “intraspecies intramuscular inoculation” mean? This sounds as if multiple species were injected into the animals. Please clarify

Line 76: This should state a case fatality rate of 62.5%

Table 1: Suggest breaking down the stray vs. pet animals in this table as well.

Line 246: Wild cats indicate non-domestic species, please clarify if the cats mentioned in this line are wild or feral domestics.

Line 249: Remove the “V” at the end of the sentence.

Line 260: Change “symptoms” to “signs”

Line 262: “Fetal” is likely a typo, should this say “fatal” instead?

Author Response

Dear Editor and reviewers,                                            Nov, 16, 2023

The file submitted here is a revised manuscript entitled “Epidemiology of Severe Fever with Thrombocytopenia Syndrome in dogs and cats in Taiwan” for your consideration to be published in the Journal “Viruses”.

Thank you very much for this opportunity to revise our manuscript, we sincerely appreciate the time the reviewers have taken to review and the very informative and insightful comments. We hope that the revised version is deemed acceptable by Viruses. The point-to-point responses to the comments are as follows, all changes are highlighted in the revised manuscript for your convenience in review. In brief, we have added more information in Methodology, one supplementary figure and several issues were included in the discussion as suggested by reviewers. The overall word count exceeds 4000 words.

Should you have any questions regarding this manuscript, please feel free to contact me using the information below. I wish to thank you in advance for your considerate attention to this manuscript and look forward to your reply.

Yours sincerely,

Chi-Chung Chou D.V.M. Ph.D.Distinguished Professor College/Department of Veterinary MedicineNational Chung-Hsing University145 Xing Da Rd. Taichung, TaiwanTEL: +886-4-22840895 ext. 519Email: ccchou@nchu.edu.tw

Wei-Li Hsu D.V.M. Ph.D.Distinguished ProfessorGraduate Institute of Microbiology and Public Health National Chung-Hsing University145 Xing Da Rd. Taichung, TaiwanTEL: +886 4 22840695Email: wlhsu@dragon.nchu.edu.tw

Round 2

Reviewer 1 Report

Comments and Suggestions for Authors

Dear authors,

Thank you vor the revision of the manuscript, you have done a lot of work, and the manuscript looks now much better.

Currently, I have just two remarks:

Lane 157-158 – “strong positive amplicons” – what do you mean here under “strong”? (Of course, I understand what do you mean, but it seems like laboratory slang). Correct, please.

Supplementary section:

The dendrogram contains bootstrap indexes lower than 75. Usually, it is advised not to show indexes less than 75. Please correct it.

Author Response

We appreciate the chance for minor revision and the comments. The manuscript has been updated accordingly. 

  1. strong positive sample was clarified in the sentence: "The identity of positive amplicons with concentrations higher than 0.1 μg/μL after purification was further confirmed".
  2. Figure S1: only the bootstrap value higher than 75 was shown on the tree. Thanks

Reviewer 2 Report

Comments and Suggestions for Authors

The authors adequately addressed my concerns/comments in the revised version of the manuscript.

Author Response

Many thanks for the recommendation.